# Sex Differences in Sleep Profiles and the Effect of Elexacaftor/Tezacaftor/Ivacaftor on Sleep Quality in Adult People with Cystic Fibrosis: A Prospective Observational Study

**DOI:** 10.3390/diagnostics14242859

**Published:** 2024-12-19

**Authors:** Sarah Dietz-Terjung, Svenja Straßburg, Tim Schulte, Paul Dietz, Gerhard Weinreich, Christian Taube, Christoph Schöbel, Matthias Welsner, Sivagurunathan Sutharsan

**Affiliations:** 1Department of Pulmonary Medicine, University Hospital Essen—Ruhrlandklinik, Adult Cystic Fibrosis Center, University of Duisburg-Essen, Tueschener Weg 40, 45329 Essen, Germany; 2Department of Sleep and Telemedicine, University Hospital Essen—Ruhrlandklinik, University of Duisburg-Essen, 45239 Essen, Germany

**Keywords:** cystic fibrosis, sleep, sleep quality, elexacaftor/tezacaftor/ivacaftor, sex differences, polysomnography

## Abstract

Background/Objectives: Recent studies indicate that sleep and sleep disorders differ between men and women, but corresponding data in people with chronic lung diseases are lacking. This study aims to answer the question of what the sex-specific differences in sleep profiles and responses to elexacaftor/tezacaftor/ivacaftor (ETI) therapy in people with cystic fibrosis (pwCF) are. Methods: Adult pwCF and a matched control group (adults with suspected sleep-disordered breathing undergoing in-laboratory polysomnography (PSG)) were included. PSG data at baseline and after 6 months’ ETI therapy were compared between men (mwCF) and women (wwCF) with cystic fibrosis. PSG data at baseline and 6-month follow-up for mwCF/wwCF were compared with baseline PSG data for men/women in the control group. Daytime sleepiness was evaluated using the Epworth Sleepiness Scale (ESS). Correlations between change in percentage predicted forced expiratory volume in 1 s from baseline to 6 months were correlated with corresponding changes in key sleep parameters. Changes in transferrin during ETI therapy were also documented. Results: Twenty-eight pwCF (12 wwCF, 16 mwCF) and 28 matched controls were included. Both mwCF (4 ± 5 vs. 9 ± 20 events/h, *p* = 0.028) and wwCF (3 ± 3 vs. 8 ± 9 events/h, *p* = 0.004) had fewer respiratory events during sleep versus male and female controls, but worse sleep efficiency (75 ± 11% vs. 84 ± 11%; *p* = 0.004 and 76 ± 10% vs. 83 ± 11%; *p* = 0.011. The baseline ESS score was significantly higher in wwCF versus female controls (8 ± 4 vs. 14 ± 8; *p* = 0.040). Although some sleep parameters normalized during ETI therapy in pwCF, sleep quality remained poor. The transferrin levels at baseline (2.7 ± 0.4 vs. 2.2 ± 0.5; *p* = 0.049) and 6 months (3.8 ± 0.4 vs. 2.6 ± 0.5; *p* < 0.001) were significantly higher in the wwCF versus the mwCF, and the change from baseline during ETI therapy was significantly greater in women versus men (1.1 ± 0.6 vs. 0.4 ± 0.4; *p* < 0.001). Conclusions: These data suggest that wwCF and mwCF should be managed differently with respect to their sleep.

## 1. Introduction

It is increasingly being recognized that sleep disorders differ between men and women. In general, women have less stage 1 sleep, but more slow wave sleep, than men [1]. Women also seem to have a greater objective sleep need and report more poor or insufficient sleep than men [2]. Furthermore, sleep in women can be affected by hormonal changes due to the menstrual cycle, pregnancy, and menopause [3,4]. Sex differences in sleep and sleep disturbances have been the subject of increasing research interest over the last decade [5,6]. However, there are few published data on sleep differences between men and women with chronic lung diseases, or on potential sex differences in the effects of pharmacological disease-specific therapy on sleep in these individuals.

Cystic fibrosis (CF) is the most common monogenic disorder. It is caused by mutations in the CF transmembrane conductance regulator (CFTR) gene on chromosome 7 [7,8]. The impact of this mutation is disturbed transport of chloride and bicarbonate ions through epithelial cell membranes, resulting in the formation of highly viscous secretions in all exocrine organs (e.g., lungs, gastrointestinal tract), which causes progressive lung damage and malnutrition [8,9,10]. People with CF (pwCF) have higher morbidity and mortality than the general population. This is mainly due to lung involvement, with progressive obstructive lung disease, hyperinflation, impaired gas exchange, and end-stage respiratory failure [11]. Advances in therapy for CF, such as triple CFTR modulatory therapy with elexacaftor/tezacaftor/ivacaftor (ETI) [10,12,13], have improved life expectancy for pwCF to more than 50 years of age, and the number of adults with CF now exceeds the number of pediatric pwCF [5,10].

There are differences between men and women with CF (mwCF and wwCF, respectively) in terms of the impact of CF, with women having lower survival rates, more lung infections, a greater decline in pulmonary function and nutritional status, and worse muscle strength/functional mobility; these differences have been attributed to the influence of estrogen [14,15,16]. This also appears to be the case during therapy for CF, with a recent study showing that the effects of ETI on several relevant clinical parameters differed between men and women [17].

Sleep-disordered breathing (SDB), mainly obstructive sleep apnea (OSA) and nocturnal hypoxemia, are common in pwCF of all ages [7,18,19,20,21]. The reported prevalence of OSA und SDB in pwCF is as high as 70% in children and up to 3.9% in adults [18,19,20]. pwCF also report poor subjective and objective sleep quality, with pronounced daytime sleepiness, often due to disease-specific issues (such as coughing or nocturnal percutaneous gastrostomy feeding), and this is associated with reduced quality of life [21]. Despite the knowledge that SDB is a common comorbidity in pwCF and that women are not only more likely to have SDB than men but also sleep differently, most CF centers do not routinely screen for SDB, especially not in a sex-specific manner [7].

This study compared sleep quality in mwCF and wwCF, before and after the initiation of ETI therapy, to determine whether treatment with ETI has a differential effect on sleep quality in men and women. Sleep-related breathing parameters and sleep quality in pwCF were also compared with those in lung-healthy individuals referred to the sleep laboratory for evaluation of SDB.

## 2. Materials and Methods

### 2.1. Study Design

This prospective, observational, and descriptive–analytical study was conducted at Ruhrlandklinik Essen, Germany, between September 2020 and March 2021. This study has ethics approval (19-8961-BO) from the Ethics Committee of the Medical Faculty of the University of Duisburg-Essen, Robert-Koch-Str. 9–11, 45147 Essen, Germany, which followed the Declaration of Helsinki Ethical Principles for Medical Research Involving Human Subjects. All included individuals provided written informed consent for participation in this study.

### 2.2. Study Participants

Participants were aged >18 years. All pwCF had a diagnosis of CF based on the presence of two defining mutations in the CFTR gene, were clinically stable without signs of respiratory exacerbation, and on stable medication, and had a stable percentage predicted forced expiratory volume in one second (ppFEV1) for ≥4 weeks before the study assessments. pwCF were matched for age and sex with lung-healthy individuals who had been referred to the sleep laboratory for evaluation of suspected SDB or sleep-related movement disorders.

### 2.3. Cardiorespiratory Polysomnography

Polysomnography (PSG) was conducted in-laboratory using a digital polygraph (Nox Medical, Reykjavík, Iceland), including EEG, EOG, EMG (submental and tibialis), rib cage and abdominal pneumograms, pulse oximeter (Nonin Medical, Inc., Minneapolis, MN, USA), and nasal cannula (with flow measurement at 20 Hz). All signals were recorded automatically and analyzed in a blinded manner by an experienced evaluator. Apnea was defined as airflow cessation for ≥10 s, and hypopnea as a ≥50% reduction in airflow or a ≥30% reduction with a >3% oxygen saturation decrease lasting ≥ 10 s. Clinically significant oxygen desaturation was SpO2 < 90% for ≥5% of total sleep time (TST). The apnea–hypopnea index (AHI) was calculated as the number of apneas and hypopneas per hour of sleep [22]. Patients with TST < 180 min were excluded.

### 2.4. Assessments

Daytime sleepiness at baseline was evaluated using the Epworth Sleepiness Scale (ESS). PSG data at baseline and at 6 months after initiation of ETI therapy were compared between mwCF and wwCF. In addition, PSG data at baseline and 6-month follow-up for mwCF and wwCF were compared with baseline PSG data for lung-healthy men and women in the control group. The latter comparisons allowed us to determine whether the findings in male and female pwCF were only sex-specific or related to CF and the effects of ETI therapy. The ppFEV1 was determined to be the most important pulmonary function parameter; correlations between the change in ppFEV1 from baseline to 6 months were correlated with changes in key sleep parameters between baseline and 6-month follow-up. Finally, changes in transferrin during ETI therapy in mwCF and wwCF were documented, and correlations with the periodic limb movement index (PLMI) were determined.

### 2.5. Laboratory Chemical Analysis

In addition, a complete blood count was taken as part of the clinical routine. Particular focus was placed on the transferrin value, as this value is relevant in sleep medicine for the diagnosis of sleep-related movement disorders.

### 2.6. Statistical Analysis

Statistical analyses were performed using the SPSS statistics package version 27 (SPSS Inc., Chicago, IL, USA). Data are presented as mean ± standard deviation. The Shapiro–Wilk test was used to evaluate the data for normality of distribution. Student’s *t*-test or Mann–Whitney U-test was used to assess between-group differences, as appropriate. The equality of variances was examined using Levene’s test. For unequal variances, the Welch test was used instead of Student’s *t*-test while the Wilcoxon test was used for equal variances. In the correlation analysis, Pearson’s correlation coefficient was calculated for parametric data, and Spearman’s correlation coefficient was used for non-parametric data. A *p*-value of <0.05 was considered statistically significant.

## 3. Results

### 3.1. Study Participants

A total of 64 pwCF were recruited, all of whom underwent full PSG; 36 were excluded from statistical analysis due to having a TST of <180 min, insufficient sleep data quality, or missing pulmonary function test data. The remaining 28 pwCF were included in the analysis (16 mwCF with a mean age of 33 ± 7 years and a mean body mass index (BMI) of 21 ± 3 (range 15.6–27.8) kg/m^2^ and 12 wwCF with a mean age of 32 ± 8 years and a mean BMI of 22 ± 3 (range 16.4–29.4) kg/m^2^; 91% of mwCF and 92% of wwCF had a mean BMI of <25 kg/m^2^). One wwCF used nocturnal oxygen supplementation, which was paused during the diagnostic PSG. No participants were using nocturnal continuous or bilevel positive airway pressure therapy.

Twenty-eight age- and sex-matched lung-healthy individuals were included in the control group (16 men with a mean age of 34 ± 6 years and a mean BMI of 34 ± 8 (range 24.1–47.8) kg/m^2^ and 12 women with a mean age of 39 ± 5 years and a mean BMI of 33 ± 5 (range 24.1–43.3) kg/m^2^).

### 3.2. Sleep Profiles of wwCF Versus mwCF

At baseline, wwCF had a significantly higher arousal index, ESS score, and nocturnal heart rate than mwCF; no other PSG parameters differed significantly between wwCF and mwCF (Table 1). The PLMI in mwCF (23 ± 12 events/h) was much higher than that in wwCF (9 ± 8 events/h), but the between-group difference did not reach statistical significance (Table 1). The baseline, AHI did not differ significantly between wwCF and mwCF (Table 1).

After 6 months of ETI therapy, the number of respiratory-effort-related arousals (RERAs) was significantly lower in wwCF versus mwCF, while the arousal index, the PLMI, the nocturnal respiration rate, and the nocturnal heat rate were significantly higher (Table 1).

### 3.3. Sleep Profiles of Men and Women with CF Versus Lung-Healthy Men and Women

Compared with lung-healthy controls, wwCF had a significantly lower AHI, oxygen desaturation index (ODI), and PLMI, and a significantly higher ESS score; differences in the AHI and ODI persisted at 6-month follow-up, when wwCF also had a significantly lower AHI during rapid eye movement (REM) sleep and sleep efficiency compared with lung-healthy women at baseline (Table 2).

For men, those with CF had a significantly lower AHI, and significantly lower TST and sleep efficiency at baseline compared with the controls (Table 2). At 6-month follow-up, mwCF had a significantly lower AHI, AHI during REM sleep, and ODI compared with the lung-healthy controls at baseline.

### 3.4. Correlations Between Changes in ppFEV1 and Changes in PSG Parameters in pwCF

There was a significant negative correlation between the change in ppFEV1 frombaseline to 6 months and the change in the AHI during REM sleep, and a significant positive correlation between the change in ppFEV1 from baseline to 6 months and the change in RERAs from baseline to 6 months in wwCF, but not mwCF; the change in ppFEV1 from baseline to 6 months did not significantly correlate with changes in any other sleep parameters during ETI therapy (Table 3).

### 3.5. Correlation Between Blood Transferrin and PLMI in pwCF

Transferrin levels at both baseline and 6 months were significantly higher in wwCF compared with mwCF, and the change from baseline during 6 months of ETI therapy was significantly greater in women versus men (Table 4). There was no significant correlation between transferrin levels and the PLMI.

## 4. Discussion

To our knowledge, this is the first study to evaluate sex differences in sleep disturbances and in the effects of ETI therapy on sleep in pwCF. Using objective PSG data, we found that wwCF had a higher arousal index, greater levels of daytime sleepiness and a higher nocturnal heart rate than mwCF. Other studies have also reported fatigue as an issue in pwCF, and this seems to be greater in wwCF versus mwCF [23,24]. Patients at our center report objectively increased fatigue with ETI therapy, which is also consistent with this study’s results. After 6 months of treatment with ETI, wwCF showed significantly more RERAs, a higher arousal index, and higher nocturnal respiratory rate and heart rate compared to mwCF. Our data suggested that ETI therapy was associated with a higher PLMI in wwCF, probably because wwCF had a significantly greater reduction in the transferrin level during ETI therapy than mwCF, indicating a ferritin reduction.

Compared with the lung-healthy controls who were undergoing PSG for the evaluation of suspected sleep disorders, both mwCF and wwCF had fewer respiratory events during sleep. At baseline, wwCF had significantly higher ESS scores than the lung-healthy controls, indicating greater daytime sleepiness. In addition, the mean baseline ESS score in wwCF was 14, well above the threshold that is considered to indicate excessive daytime sleepiness (>10). After 6 months of ETI therapy, wwCF had a PLMI that was equal to the baseline value in lung-healthy women and significantly worse sleep quality. Lower sleep efficiency may reflect more sleep disturbance due to the greater number of PLMI during ETI therapy. Overall, mwCF had better sleep than the lung-healthy male controls from a respiratory event perspective, but did have worse sleep quality and a shorter total sleep time at baseline.

Our results are in line with other current studies. In their review, Reiter et al. [25] noted that pediatric pwCF experience disturbed sleep due to SDB and nocturnal hypoxemia. Additionally, Vandeleur et al. [26] reported a strong association between the occurrence of SDB and the respiratory outcome, which we were able to confirm in our analyses. Cohen-Cymberknoh and colleagues [27] investigated the sleep of adult patients with primary ciliary dyskinesia and CF and its correlation with quality of life. They found that sleep disorders and bad sleep quality were common in these individuals. Additionally, sleep quality appeared to be associated with disease severity and had an impact on quality of life. We also documented a connection between illness severity and sleep quality. In addition, we found that the iron-related parameters, which show overall improvement during ETI therapy in pwCF, may actually worsen in wwCF.

Overall, it can be assumed that ETI, by enhancing CFTR function, might influence several systems in the body, including the brain, which controls sleep patterns [7,28]. In pwCF, where the CFTR function is likely closer to normal but still impacted by the mutations, ETI could enhance CFTR function to a greater extent compared to pwCF with more severely impaired CFTR activity.

Speculatively, ETI could affect sleep through multiple pathways. One of the prominent consequences of CFTR dysfunction is its impact on the respiratory system [28,29,30,31,32,33,34]. In pwCF, poor lung function can lead to sleep disturbances, such as sleep apnea or poor sleep quality due to difficulty in breathing [7]. By improving lung function, ETI could help alleviate these respiratory-related sleep disturbances. This would be more noticeable in patients who have more severe lung issues due to their reduced CFTR function.

Additionally, the regulation of circadian rhythms can be influenced by various biological processes, including ion transport, which is also regulated by CFTR [28,30]. If ETI improves CFTR function in the brain, it could theoretically enhance circadian rhythm regulation, potentially leading to improved sleep quality.

Last, chronic inflammation is common in CF, particularly in the lungs, but also systemically. This inflammation may extend to the central nervous system, contributing to sleep disturbances. ETI’s anti-inflammatory effects could reduce neuroinflammation, leading to better sleep [32,33,34].

Furthermore, CFTR expression is not only present in the lungs and gastrointestinal tract but is also found in the brain, albeit at lower levels [29,31]. It has been suggested that CFTR might have a role in regulating neural processes, including the control of sleep and cognition. The blood–brain barrier (BBB) complicates direct drug delivery to the central nervous system (CNS), but recent studies [29,31] suggest that CFTR modulators, such as ETI, might have an indirect impact on brain function by improving overall CFTR function in peripheral tissues, which could influence brain signaling pathways.

It is plausible that ETI might influence CFTR in the brain to some extent, although research specifically linking CFTR modulators with brain function is still emerging. By improving the overall CFTR function in the body [32,34], ETI may indirectly modulate brain chemistry, affecting neurotransmitter release, neuronal signaling, and potentially improving sleep regulation. However, definitive evidence of a direct effect on CFTR in the brain is still limited.

Transferrin is a glycoprotein that binds and transports iron in the blood. There is some evidence to suggest that CFTR dysfunction might affect iron homeostasis, and, indirectly, transferrin levels [28,30]. The primary mechanisms through which CFTR might influence transferrin levels could involve the regulation of iron transport across epithelial membranes, which could affect systemic iron distribution, which could explain our findings.

In CF, iron absorption is already compromised due to gastrointestinal malabsorption, and the chronic inflammation associated with the disease might also alter iron metabolism. CFTR modulators like ETI might improve this dysfunction by restoring more normal iron absorption or altering systemic inflammation. While there is no direct evidence that CFTR modulates transferrin levels specifically, it might be plausible that by improving CFTR function, ETI could have a secondary impact on iron metabolism and transferrin levels.

To sum up, ETI appears to have complex, multi-faceted effects on sleep parameters in CF, probably influenced by the degree of CFTR dysfunction. While the direct effects of CFTR in the brain remain under investigation, it seems plausible that ETI may influence brain function indirectly by enhancing CFTR function in peripheral tissues, possibly impacting sleep and cognitive functions. Further research is needed to clarify these mechanisms and their implications for CF treatment.

A key strength of this study is its novelty in evaluating sex differences in sleep in pwCF, and in the effects of ETI therapy on sleep. However, some limitations also need to be acknowledged. First, the sample size was small, with only twelve wwCF and sixteen mwCF presenting for repeat sleep laboratory examination at the 6-month follow-up. In addition, pwCF did not complete the ESS at the 6-month follow-up, so we were unable to determine the impact of ETI on daytime sleepiness, both overall and in mwCF versus wwCF. A 2-year follow-up is planned, which will include the ESS, and will hopefully also include more pwCF undergoing repeat PSG

## 5. Conclusions

Our results confirm those of other recent studies recommending that pwCF undergo regular sleep evaluations. However, the current findings also suggest that wwCF and mwCF should be managed differently with respect to their sleep. For women, the focus should be on PLMI, sleep quality, and transferrin levels, while in men, the focus should be on the AHI, as well as total sleep time and sleep efficiency, which means the sleep quality. This will allow a more personalized approach to comprehensive patient management that should help pwCF to achieve a good quality of life.

## Figures and Tables

**Table 1 diagnostics-14-02859-t001:** Polysomnographic data for women and men with cystic fibrosis at baseline and after 6 months of elexacaftor/tezacaftor/ivacaftor therapy. Significant *p*-values are marked as bold.

	Baseline			6-Month Follow-Up		
	wwCF (*n* = 12)	mwCF (*n* = 16)	*p*-Value	wwCF (*n* = 12)	mwCF (*n* = 16)	*p*-Value
AHI, events/h	3 ± 3	4 ± 5	0.199	1 ± 1	2 ± 2	0.169
Supine AHI, events/h	9 ± 10	7 ± 14	0.342	1 ± 1	9 ± 8	0.269
AHI during REM sleep, events/h	8 ± 12	10 ± 10	0.499	1 ± 1	3 ± 1	0.377
RERAs, events/h	18 ± 22	10 ± 9	0.499	1 ± 1	16 ± 1	**0.049**
ODI, events/h	6 ± 7	8 ± 5	0.352	2 ± 1	8 ± 2	0.246
Arousal index, events/h	36 ± 65	16 ± 12	**0.049**	19 ± 5	14 ± 5	**0.049**
Total sleep time, min	291 ± 39	301 ± 42	0.267	309 ± 32	309 ± 9	0.281
Sleep efficiency, %	76 ± 10	75 ± 11	0.449	67 ± 6	72 ± 2	0.205
Sleep latency, min	76 ± 10	75 ± 11	0.449	109 ± 30	73 ± 11	0.302
Wake time after sleep onset, min	28 ± 21	28 ± 19	0.407	40 ± 11	44 ± 15	0.407
Time spent in N1 sleep, min	6 ± 4	9 ± 4	0.07	6 ± 2	9 ± 1	0.06
Time spent in N2 sleep, min	159 ± 29	160 ± 26	0.458	158 ± 10	159 ± 7	0.458
Time spent in N3 sleep, min	78 ± 26	77 ± 22	0.468	81 ± 11	73 ± 5	0.469
Time spent in REM sleep, min	55 ± 27	53 ± 22	0.409	53 ± 11	48 ± 4	0.412
Time spent awake, min	87 ± 33	98 ± 46	0.263	88 ± 16	103 ± 12	0.254
PLMI, events/h	9 ± 8	23 ± 12	0.057	48 ± 32	19 ± 6	**0.013**
ESS score	14 ± 8	6 ± 3	**0.049**	-	-	-
Mean nocturnal SpO_2_, %	93 ± 2	92 ± 2	0.454	95 ± 1	94 ± 2	0.454
Minimum nocturnal SpO_2_, %	87 ± 2	88 ± 4	0.418	90 ± 1	88 ± 1	0.417
Nocturnal respiration rate, breaths/min	22 ± 4	21 ± 4	0.109	17 ± 1	15 ± 1	**0.033**
Nocturnal heart rate, beats/min	74 ± 12	60 ± 8	**0.001**	68 ± 5	57 ± 3	**0.007**

Values are mean ± standard deviation; ESS, Epworth Sleepiness Scale; mwCF, men with cystic fibrosis; N1/2/3, non-rapid-eye-movement sleep stage 1/2/3; ODI, oxygen desaturation index; PLMI, periodic leg movements index; REM, rapid eye movement; RERAs, respiratory-effort-related arousals; SpO_2_, oxygen saturation; wwCF, women with cystic fibrosis.

**Table 2 diagnostics-14-02859-t002:** Polysomnographic data for men and women with cystic fibrosis at baseline and after 6 months of elexacaftor/tezacaftor/ivacaftor therapy compared with baseline values for male and female lung-healthy controls referred for evaluation of suspected sleep disorders. Significant *p*-values are marked as bold.

	Controls (Baseline)	Cystic Fibrosis (Baseline)	*p*-Value vs. Controls	Cystic Fibrosis (6-Month Follow-Up)	*p*-Value vs. Controls at Baseline
**Women**	(*n* = 12)	(*n* = 12)		(*n* = 12)	
AHI, events/h	8 ± 9	3 ± 3	**0.004**	1 ± 1	**0.004**
AHI during REM sleep, events/h	10 ± 10	8 ± 12	0.068	1 ± 1	**0.004**
ODI, events/h	12 ± 14	6 ± 7	**0.023**	2 ± 1	**0.023**
Total sleep time, min	336 ± 73	291 ± 39	0.316	309 ± 32	0.074
Sleep efficiency, %	83 ± 11	76 ± 10	0.091	67 ± 6	**0.011**
PLMI, events/h	33 ± 37	9 ± 8	**<0.001**	48 ± 33	0.566
ESS score	8 ± 4	14 ± 8	**0.040**	-	-
**Men**	(*n* = 16)	(*n* = 16)		(*n* = 16)	
AHI, events/h	9 ± 20	4 ± 5	**0.028**	2 ± 2	**0.018**
AHI during REM sleep, events/h	12 ± 20	10 ± 10	0.068	3 ± 1	**0.049**
ODI, events/h	11 ± 5	8 ± 5	0.319	2 ± 1	**0.014**
Total sleep time, min	345 ± 42	301 ± 42	**0.045**	309 ± 9	0.143
Sleep efficiency, %	84 ± 11	75 ± 11	**0.020**	72 ± 2	0.118
PLMI, events/h	29 ± 32	23 ± 12	0.319	19 ± 6	0.068
ESS score	6 ± 4	6 ± 3	0.783	-	-

Values are mean ± standard deviation. AHI, apnea–hypopnea index; ESS, Epworth Sleepiness Scale; ODI, oxygen desaturation index; PLMI, periodic leg movements index; REM, rapid eye movement.

**Table 3 diagnostics-14-02859-t003:** Correlations between the change in percent predicted forced expiratory volume in 1 s and the change in polysomnography parameters after 6 months of elexacaftor/tezacaftor/ivacaftor therapy in men and women with cystic fibrosis. Significant *p*-values are marked as bold.

	wwCF (*n* = 12)	mwCF (*n* = 16)
**Correlations with the change in ppFEV_1_ from baseline to 6 months**		
Change in AHI from baseline to 6 months	–0.071 (0.827)	0.114 (0.686)
Change in supine AHI from baseline to 6 months	–0.123 (0.703)	0.013 (0.965)
Change in AHI during REM sleep from baseline to 6 months	**–0.638 (0.025)**	0.072 (0.798)
Change in RERAs from baseline to 6 months	**0.609 (0.035)**	0.179 (0.508)
Change in ODI from baseline to 6 months	–0.439 (0.177)	–0.119 (0.685)
Change in arousal index from baseline to 6 months	0.528 (0.095)	–0.184 (0.513)
Change in total sleep time from baseline to 6 months	0.217 (0.498)	–0.04 (0.990)
Change in sleep efficiency from baseline to 6 months	0.046 (0.888)	0.113 (0.689)
Change in sleep latency from baseline to 6 months	0.228 (0.477)	–0.143 (0.612)
Change in wake time after sleep onset from baseline to 6 months	–0.365(0.300)	0.270 (0.350)
Change in nocturnal mean SpO_2_ from baseline to 6 months	0.088 (0.786)	–0.350 (0.201)
Change in nocturnal minimum SpO_2_ from baseline to 6 months	0.353 (0.288)	–0.459 (0.085)
Change in nocturnal respiration rate from baseline to 6 months	0.518 (0.084)	0.222 (0.427)
Change in nocturnal heart rate from baseline to 6 months	0.049 (0.880)	0.450 (0.092)

Data are r (*p*-value). AHI, apnea–hypopnea index; mwCF, men with cystic fibrosis; ODI, oxygen desaturation index; REM, rapid eye movement; RERAs, respiratory-effort-related arousals; SpO_2_, oxygen saturation; wwCF, women with cystic fibrosis.

**Table 4 diagnostics-14-02859-t004:** Transferrin levels for men and women with cystic fibrosis at baseline and after 6 months of elexacaftor/tezacaftor/ivacaftor therapy. Significant *p*-values are marked as bold.

	wwCF (*n* = 12)	mwCF (*n* = 26)	*p*-Value
**Transferrin, g/L**			
Baseline	2.7 ± 0.4	2.2 ± 0.5	**0.0049**
6-month follow-up	3.8 ± 0.4	2.6 ± 0.5	**<0.001**
Change from baseline to 6 months	1.1 ± 0.6	0.4 ± 0.4	**<0.001**

Values are mean ± standard deviation. mwCF, men with cystic fibrosis; wwCF, women with cystic fibrosis.

## Data Availability

Data are unavailable due to privacy or ethical restrictions.

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
