# Peer review of "Sex Differences in Sleep Profiles and the Effect of Elexacaftor/Tezacaftor/Ivacaftor on Sleep Quality in Adult People with Cystic Fibrosis: A Prospective Observational Study"

_diagnostics, 2024, doi:10.3390/diagnostics14242859_

Round 1

Reviewer 1 Report

Comments and Suggestions for Authors

The manuscript by Dietz-Terjung et al examines sleep difference between wwCF and mwCF at baseline and 6 months post initiation of ETI treatment.  This approach is novel in that men and women have different sleep patterns and likely different responses to ETI regarding sleep quality.  The overall premise of the paper is sound and the results are interesting in that ETI does not fix all aspects of sleep in CF. 

Comments

1.      The authors analyze significance between wwCF and mwCF pre- and post-ETI therapy compared to controls.  The significance of changes in response to ETI is never directly examined between wwCF, pre- v. wwCF, post-ETI and the same for mwCF.  For example, supine AHI decreases significantly for wwCF post-ETI but not for men.  The analysis of male and female responses is never compared directly and some interesting difference appear to be present. 

2.     Line 147, BMI is listed as a mean of 34 + 8 with a range of 24.1 to 27.8.

3.     More explanation of why transferrin was measured needs to be included.  There is not mention in the introduction or results that it is measured to correlate with limb movement.  It is not clear why it was measured until the discussion.

4.     The authors never discuss or speculate as to how ETI is having these effects on sleep parameters in wwCF v mwCF.  Is ETI effecting CFTR in the brain?  Does CFTR influence transferrin levels?  A broader discussion of the impact of ETI would enhance the paper.

Author Response

Changes in manuscript no. diagnostics-3268653 “Sex differences in sleep profiles and the effect of elexacaftor/tezacaftor/ivacaftor on sleep quality in adult people with cystic fibrosis: a prospective observational study”.

Authors: Sarah Dietz-Terjung, Svenja Strassburg, Tim Schulte, Paul Dietz, Gerhard Weinreich, Christian Taube, Christoph Schöbel, Matthias Welsner, Sivagurunathan Sutharsan

Remarks reviewer #1

Changes/replies

1. The authors analyze significance between wwCF and mwCF pre- and post-ETI therapy compared to controls.  The significance of changes in response to ETI is never directly examined between wwCF, pre- v. wwCF, post-ETI and the same for mwCF.  For example, supine AHI decreases significantly for wwCF post-ETI but not for men.  The analysis of male and female responses is never compared directly and some interesting difference appear to be present. 

1.    Thank you for your astute review and very good advice.  A more in-depth analysis with a larger patient group is currently being worked on in the form of a doctoral thesis at our company and we look forward to discussing these results soon.

2. Line 147, BMI is listed as a mean of 34 + 8 with a range of 24.1 to 27.8

2. Thank you for this, we corrected this value.

3. More explanation of why transferrin was measured needs to be included.  There is not mention in the introduction or results that it is measured to correlate with limb movement.  It is not clear why it was measured until the discussion

3. We have added to the methods section (lines 125-129):

„2.5 LABORATORY CHEMICAL ANALYSIS.

In addition, a complete blood count was taken as part of the clinical routine. Particular focus was placed on the transferrin value, as this value is relevant in sleep medicine for the diagnosis of sleep-related movement disorders. „  

4. The authors never discuss or speculate as to how ETI is having these effects on sleep parameters in wwCF v mwCF.  Is ETI effecting CFTR in the brain?  Does CFTR influence transferrin levels?  A broader discussion of the impact of ETI would enhance the paper.

4. We thank you for this advice. We have added to the discussion from line 244 and tried to address your comments accordingly.

Reviewer 2 Report

Comments and Suggestions for Authors

As a general comment I believe that the control group should be mathced with the pwCF for BMI too. The sleep differences between the two groups could be probably attributed to the differences in BMI. Therefore, I believe that the inclusion of the specific control group in the present form does not add anything to the paper. Perhaps, the authors will include a control group matched for BMI in their future paper that they plan to write with the 2 year follow-up.

In the methods, the "CARDIORESPIRATORY POLYSOMNOGRAPHY" section is very detailed. Since the authors cite the corresponding references, I believe that they should not describe polysomnography or scoring practices with that much detail.

In my opinion the "Discussion" section of the manuscript is rather short and could be improved with more elaboration and references.

In the "Conclusions" section of the manuscript the authors write that "in men the focus should be on the AHI and ODI". In my opinion, this is a conclusion that is not supported by the results.

Comments on the Quality of English Language

Although I am not a native English speaker I found several minor linguistic issues in the manuscript. These issues might not limit the understanding of the manuscript significantly, however, I think that the quality of the manuscript will be improved if these issues get adressed.

Author Response

Changes in manuscript no. diagnostics-3268653 “Sex differences in sleep profiles and the effect of elexacaftor/tezacaftor/ivacaftor on sleep quality in adult people with cystic fibrosis: a prospective observational study”.

Authors: Sarah Dietz-Terjung, Svenja Strassburg, Tim Schulte, Paul Dietz, Gerhard Weinreich, Christian Taube, Christoph Schöbel, Matthias Welsner, Sivagurunathan Sutharsan

Remarks reviewer # 2

Changes/replies

1. As a general comment I believe that the control group should be mathced with the pwCF for BMI too. The sleep differences between the two groups could be probably attributed to the differences in BMI. Therefore, I believe that the inclusion of the specific control group in the present form does not add anything to the paper. Perhaps, the authors will include a control group matched for BMI in their future paper that they plan to write with the 2 year follow-up.

1. Many thanks for your efforts and the very good and helpful review.

You are absolutely right. Unfortunately, it was very difficult to match the BMI, as some of our pwCF had a very low BMI at the beginning of the study. We did try not to make the difference too big when matching, but we don't want to call it matched.

However, we have a larger group of patients in a doctoral thesis we are currently working on and can include this in the study.

1.    In the methods, the "CARDIORESPIRATORY POLYSOMNOGRAPHY" section is very detailed. Since the authors cite the corresponding references, I believe that they should not describe polysomnography or scoring practices with that much detail.

2.    Thank you for your feedback, we have shortened this section to (line 98-107):

“Polysomnography (PSG) was conducted in-laboratory using a digital polygraph (Nox Medical, Iceland), including EEG, EOG, EMG (submental and tibialis), rib cage and abdominal pneumograms, pulse oximeter (Nonin, Minnesota, USA), and nasal cannula (with flow measurement at 20 Hz). All signals were recorded automatically and analyzed in a blinded manner by an experienced evaluator. Apnea was defined as airflow cessation for ≥10 seconds, and hypopnea as a ≥50% reduction in airflow or a ≥30% reduction with a >3% oxygen saturation decrease lasting ≥10 seconds. Clinically significant oxygen desaturation was SpO2 <90% for ≥5% of total sleep time (TST). The apnea-hypopnea index (AHI) was calculated as the number of apneas and hypopneas per hour of sleep. Patients with TST <180 minutes were excluded.“

3. In my opinion the "Discussion" section of the manuscript is rather short and could be improved with more elaboration and references.

3. We thank you for this advice. We have added to the discussion from line 244 and tried to address your comment accordingly.

4. In the "Conclusions" section of the manuscript the authors write that "in men the focus should be on the AHI and ODI". In my opinion, this is a conclusion that is not supported by the results.

4. We thank you very much for your efforts. We changed the conclusion to “. For women, the focus should be on PLMI, sleep quality and transferrin levels, while in men the focus should be on the AHI as well as Total Sleep Time and Sleep efficiency, which means the sleep quality. „ and hope that this will meet with your approval.

5. Although I am not a native English speaker I found several minor linguistic issues in the manuscript. These issues might not limit the understanding of the manuscript significantly, however, I think that the quality of the manuscript will be improved if these issues get adressed.

5. We have checked the text again and thank you for pointing this out.

Round 2

Reviewer 2 Report

Comments and Suggestions for Authors

Thank you for the changes that you made on your manuscript based on my suggestions.

I wish you good luck on your next research.

Congratulations!!!

Author Response

Thank you so much!